

# Seasonality of the North Pacific Ocean Desert area in the past two decades and a modelling perspective for the 21st century

Siyu Meng[1,2,3], Xun Gong[4,5,6], Benjamin Webber[3], Manoj Joshi[3], Xiaokun Ding[7], Xiang Gong[8,9], Mingliang Gu[1,2], Huiwang Gao[1,2]

[1]Frontiers Science Center for Deep Ocean Multispheres and Earth System, and Key Laboratory of Marine Environment and Ecology, Ministry of Education of China, Ocean University of China, Qingdao, China.
[2]Laboratory for Marine Ecology and Environmental Sciences, Laoshan Laboratory for Marine Science and Technology, Qingdao, China.
[3]Climatic Research Unit, School of Environmental Sciences, University of East Anglia, Norwich, United Kingdom.
[4]Institute for Advanced Marine Research, China University of Geosciences, Guangzhou, China.
[5]State Key Laboratory of Biogeology and Environmental Geology, Hubei Key Laboratory of Marine Geological Resources, China University of Geosciences, Wuhan, China.
[6]Shandong Provincial Key Laboratory of Computer Networks, Qilu University of Technology (Shandong Academy of Sciences), Jinan, China.
[7]School of Ocean, Yantai University, Yantai, China
[8]School of Mathematics and Physics, Qingdao University of Science and Technology, Qingdao, China
[9]Qingdao Innovation Center of Artificial Intelligence Ocean Technology, Qingdao, China

*Correspondence to*: Xun Gong (gongxun@cug.edu.cn) and Huiwang Gao (hwgao@ouc.edu.cn)

## Abstract

As the largest ocean desert, the North Pacific Ocean Desert (NPOD) exhibits pronounced variations across seasonal, decadal, and centennial time scales. Notably, changes in the seasonality of the NPOD are thought to have larger effects on marine ecosystems than variability in the annual-mean state of the NPOD. However, the interannual variability of NPOD seasonality and its response to climate processes remain unclear. Here, we investigate the amplitude of the seasonal cycle in NPOD area and its linkage with climate variability and change. Our results show that the El Niño - Southern Oscillation (ENSO)

modulated the seasonal maximum of NPOD area in boreal summer, and thus the amplitude of the seasonal cycle during 1998–2021. This is primarily due to ENSO-induced changes in nutrient transport via equatorial upwelling and thermal stratification. Future projections based on Coupled Model Intercomparison Project Phase 5 (CMIP5) modelling results and an Elman neural network indicate a significant decrease in the seasonal amplitude of NPOD area by 2100, attributed to the growing seasonal minimum of NPOD area in winter along the anthropogenic increase in atmospheric $CO_2$. The findings

highlight the importance of considering seasonal differences in future research on the interannual variability of ocean desert and underscore the need for models to distinguish between the effects of climate variability and change.



## 1 Introduction

Ocean deserts are the regions of low surface Chlorophyll-a (Chl-a) concentrations (less than 0.07 mg m$^{-3}$), occupying approximately 40% of the global ocean, and are mainly found at subtropical latitudes (McClain et al., 2004). As the largest

ocean desert, the North Pacific Ocean Desert (NPOD) exhibits variability in area, intensity and location (Meng et al., 2021), coherent with climate dynamics on seasonal (Signorini et al., 2015), decadal (Signorini and McClain, 2012) and centennial (Boyce et al., 2014) time scales. Studies have suggested that the variations of the seasonal cycle in Chl-a and primary production may have a larger impact on the survival of marine species, and hence the oceanic food web and the ocean carbon cycle, than changes in their annual mean quantities (Lutz et al., 2007; Muñiz et al., 2021), suggesting a need to better

quantify any changes in the seasonal cycle of the NPOD.

In previous studies, the NPOD has been quantified by its area and intensity, i.e. spatially averaged Chl-a concentration, on the basis of satellite-derived ocean colour datasets (Leonelli et al., 2022; Meng et al., 2021; Wilson and Qiu, 2008), in situ data (Gregg and Rousseaux, 2014), and research cruise observations (Raes et al., 2022). Using satellite observations, McClain et al. (2004) have suggested synchronous variations in the NPOD intensity and area on seasonal and interannual

time scales, indicating that increased NPOD intensity usually corresponds to expanded NPOD area. However, extended periods of satellite observation have prompted numerous studies to propose a more pronounced increase in NPOD area than intensity in response to climate processes (Irwin and Oliver, 2009; Meng et al., 2021). Therefore, this study primarily focuses on the variations of NPOD area.

In particular, the seasonal cycle of NPOD area has been linked to nutrient availability in the surface ocean (Behrenfeld et al.,

2006; Henson et al., 2013; Kwiatkowski et al., 2017). In boreal summer, higher sea surface temperatures (SSTs) inhibit the mixing between surface water and the subsurface nutrient-rich water, by enhancing vertical stratification and shoaling the mixed layer depth (MLD) (Signorini et al., 2015). As a result, the reduced availability of nutrients in the surface waters limits the growth of phytoplankton, leading to lower Chl-a concentrations and NPOD expansion in summer. Conversely, in winter, the mixed layer deepens, due to relatively low SSTs, entraining nutrient-rich water and causing chlorophyll blooms

(Mao et al., 2020).

Studies have suggested that the variation of Chl-a concentration in subtropical ocean is likely driven by basin-wide climate processes. For example, the El Niño - Southern Oscillation (ENSO), characterized by the variability over 2–7 years, can regulate the nutrient transfer to the upper ocean via changes in ocean horizontal advection and upwelling (Racault et al., 2017). Over longer timescales, the Pacific Decadal Oscillation (PDO) reflects variability in the patterns of ocean circulation

and SSTs, which are associated with changes in the nutrients availability and Chl-a concentrations in NPOD (Martinez et al., 2009; Meng et al., 2021). In parallel, climate warming has been shown to intensify the thermal stratification and nutrient limitation in the upper ocean and consequently leads to NPOD expansion on a time scale of several decades (Henson et al., 2010; Lewandowska et al., 2014; Meng et al., 2021). Hence, to achieve a comprehensive understanding of how NPOD





seasonality responds to the dual influences of interannual climate variability and long-term climate change, it is crucial to
investigate the NPOD seasonality across different time scales.

While previous research has acknowledged the potential effects of phytoplankton seasonality on marine ecosystems and the
carbon cycle, the variation in the seasonality of NPOD, the ocean with the lowest phytoplankton level globally, and its
response to climate processes remain unclear. In this study, we aim to better understand and quantify the variation in the
amplitude of the seasonal cycle (i.e. the difference between maximum and minimum in the annual cycle) in NPOD area. To
do this, we use satellite Chl-a data during 1998–2021, and also evaluate future projections for the 21st century based on an
Elman Neural Network (ENN) model and results from the Coupled Model Intercomparison Project Phase 5 (CMIP5) (Taylor
et al., 2012).








## 2 Data and Methods

### 2.1 Data

In this study, the NPOD is defined as the region within which the ocean surface Chl-a concentration is less than 0.07 mg m$^{-3}$, following McClain et al. (2004) and Polovina et al. (2008). We obtain Chl-a concentration data from Sea-viewing Wide Field-of-View Sensor (SeaWiFS) and Moderate Resolution Imaging Spectroradiometer (MODIS/Aqua) ocean colour observations (NASA OBPG, 2014). Here, a 13 day running-mean and 11 grids spatial smoothing are utilized to fill in the missing values of Chl-a data due to cloud coverage (Cole et al., 2012). Then, the SeaWiFS dataset from 1998 to 2007 and

MODIS dataset from 2003 to 2021 are merged to form a continuous dataset of monthly Chl-a observation at 9 km spatial resolution from 1998 to 2021. To achieve this, a cross-calibration is performed at each grid point based on the overlapping period in the SeaWiFS and MODIS-Aqua datasets (Fig. S1). Overall, there are 1719140 pairs of concurrent Chl-a concentration data in NPOD region over the 5 years overlapping period, and the two datasets present a R-squared coefficient of 0.78 with offset of 0.0025, suggesting a strong coherence between the SeaWiFS and the MODIS datasets (Fig. S1). The

merged database has been validated in Meng et al. (2021) and used to analyse decadal variability in the NPOD.

  All observational and reanalysis data used in this study are listed in Table S1. Specifically, the Optimum Interpolation SST data from National Oceanic and Atmospheric Administration (NOAA) (Reynolds, 1988), sea surface height data from Global Ocean Data Assimilation System provided by NOAA Physical Sciences Division, MLD data from the Simple Ocean Data Assimilation version 3 (SODA3) reanalysis dataset (Carton et al., 2018), downward solar radiation and precipitation

rate data from European Center for Medium Range Weather Forecasting (ECMWF) ERA5 reanalysis dataset (Hersbach et al., 2019), and wind stress data from ECMWF ORAS5 (Zuo et al., 2019) are used to explore the response of NPOD seasonality to physical climate variability. Wind stress curl is calculated from the ORAS5 wind stress to investigate the role of wind-driven downwelling in the NPOD. Furthermore, ocean horizontal velocity components, temperature and salinity data from the SODA3 reanalysis data from 1998 to 2015 are used to force a one-dimensional K-profile parameterization

ocean model (Large et al., 1994) to evaluate ocean vertical mixing process at each grid point in the NPOD area. Nutrient concentration data (Nitrate + Phosphate) from World Ocean Atlas 2005 (Levitus, 2006) is combined with K-profile parameterization analysis to calculate nutrient fluxes across 10 m depth into the surface ocean (Text S1). The ENSO signal is represented by the Niño 3.4 index, which is obtained from NOAA Physical Sciences Laboratory (Rayner et al., 2003) and calculated by using the HadISST1 dataset.

For the projections of the NPOD area in the 21st century, we apply the ENN machine learning model to 7 CMIP5 simulations (Table S2) based on the Representative Concentration Pathway 8.5 (RCP 8.5) scenario (Taylor et al., 2012). RCP 8.5 experiment simulates a climate change scenario which is forced by prescribed greenhouse gas and other natural forcings, and the radiative forcing value is projected to rise to 8.5 W m$^{-2}$ by 2100. It provides an insight into the climate impacts of high-end greenhouse gas emission pathways (Moss et al., 2010; Schwalm et al., 2020). The CMIP5 modelling



outputs provide data at monthly resolution from 2006 to 2100. These outputs are standardized by converting them to Z-scores before being input into the ENN model. Further description of the CMIP5 data is provided in Text S2.

## 2.2 Elman Neural Network

A key aim of this study is to project the NPOD area seasonality in a future climate change scenario. However, the overall underestimation of Chl-a concentration in subtropical gyres in Earth System Models (ESMs) (Séférian et al., 2013) precludes

using such models to identify the boundary and area of the NPOD (Fig. S2c). Compared with Chl-a concentration, the physical properties of the ocean and atmosphere that are key factors regulating the NPOD variation, e.g. SST and wind stress curl, are better represented in the NPOD region (Fig. S3). Therefore, an ENN model is combined with physical variables from CMIP5 modelling outputs (models listed in Table S2 and more details in Text S2) to make projections of NPOD area over the 21st century. ENN (Elman, 1990) is a typical recurrent neural network which reuses past information as inputs to

predict the next or future states. Compared to traditional neural networks consisting of input, hidden and output layers, ENN adds a context layer to pass the information from the last network iteration to the current iteration. Thus, ENN is more suitable to model temporal sequences especially with strong periodic variations like the seasonality of the NPOD area.

Although ENN is an effective method to predict time series, when NPOD area variation exhibits a high degree of nonstationary due to the dual effects of climate variability and change, the accuracy and robustness of ENN may be reduced

(Stock et al., 2018). Therefore, we use the time series of SST, wind stress curl and solar radiation in NPOD region, based on the observation and reanalysis data, as the input data of ENN, and the time series of NPOD area as the output data to evaluate the ENN performance. These input variables are selected by a sensitivity test that assesses ENN performance with different input configurations (Table S3). Here, 69% of the NPOD area time series is used for ENN training, and 31% of the NPOD area time series is used to verify the error between the NPOD areas predicted by ENN and observed by satellite. As

shown in Fig. S4, the relative error of ENN prediction is only 7.06%, indicating the good performance of ENN in projecting NPOD area (Mean Absolute Error = 1.75 $10^6$ km$^2$, Root Mean Square Error = 2.46 $10^6$ km$^2$, R-squared = 0.82). Moreover, the ENN results are compared with the NPOD area simulations by three ESMs, CMCC-CESM, CNRM-CM5 and IPSL-CM5A, which are best able to capture observed Chl-a distribution among CMIP5 models (Fu et al., 2022). Although the seasonal cycles of NPOD area are well simulated in both ESMs and ENN (Figs. S2b, S4), the systematic biases of NPOD

area and location in ESMs are very substantial, with the average simulated NPOD area being more than double the observed area (Fig. S2a, c).




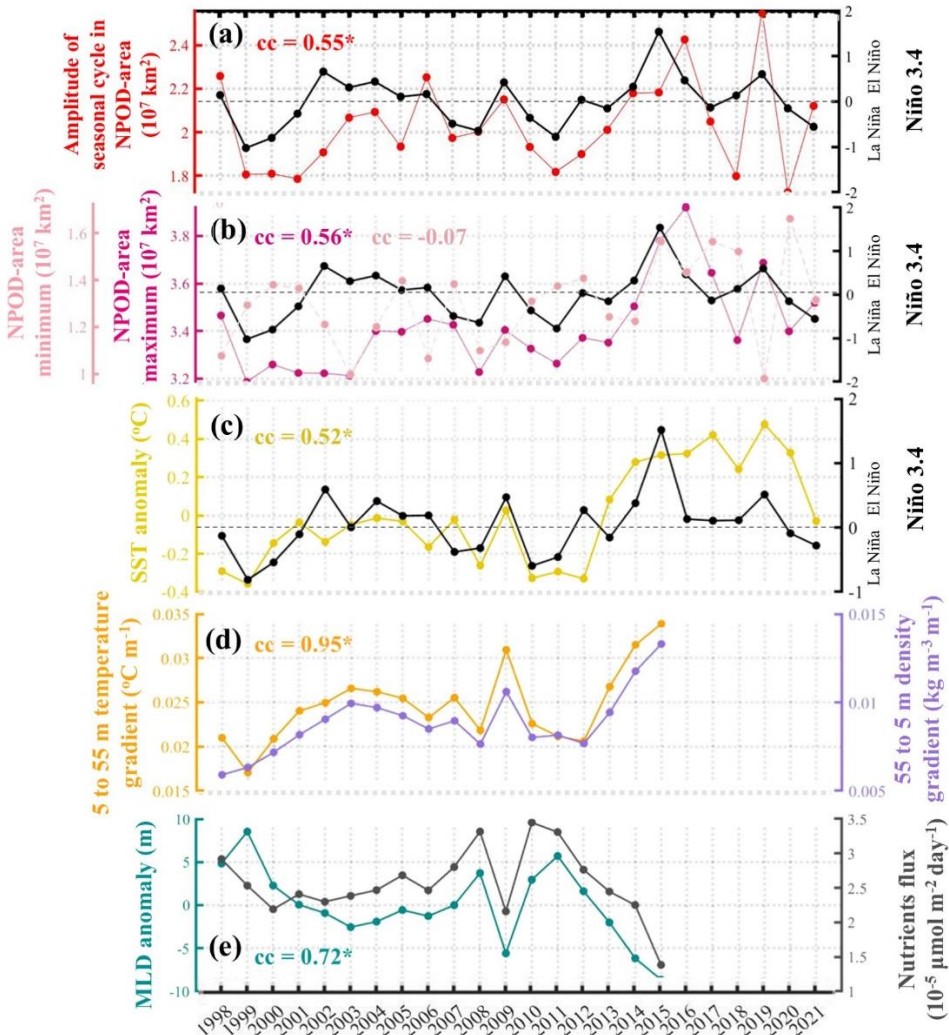

**Figure 1. Interannual variation in NPOD area seasonal cycle as a response to ENSO. (a) Amplitude of the seasonal cycle in NPOD area (left axis) and Niño 3.4 index (right axis) in 1998–2021. (b) NPOD area seasonal maximum (dark red, solid) and minimum (light red, dashed) and Niño 3.4 index (black) in 1998–2021. (c-e) SST anomaly in NPOD and Niño 3.4 index (c), temperature gradient and density gradient of 5 to 55 m (d), MLD anomaly and upward nutrient flux from vertical mixing (e) in NPOD in each summer half year. "cc" represents the correlation coefficient, "*" represents the p-value less than 0.01. All variables in (c-e) are averaged within the fixed NPOD region, which is determined by the multi-year average of Chl-a data.**





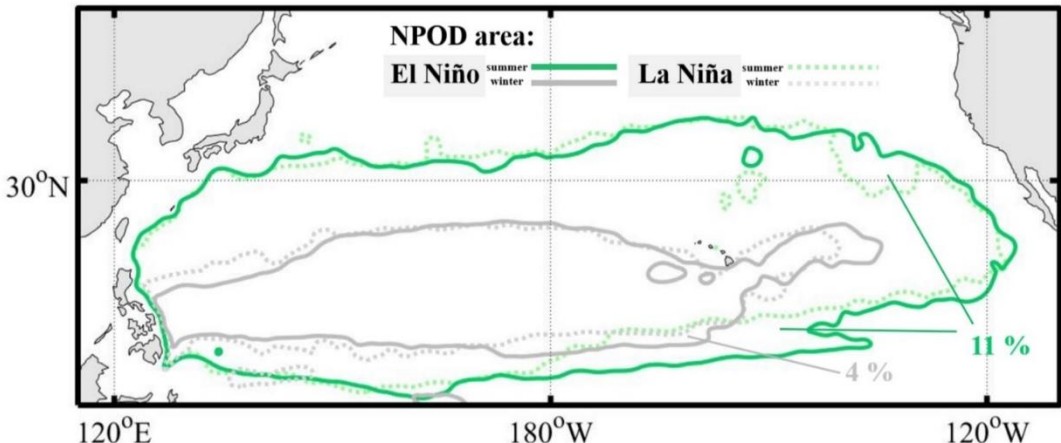

**Figure 2. The boundaries of NPOD seasonal maximum area (green lines) and minimum area (grey lines) in El Niño (dark colour, solid) and La Niña (light colour, dashed) years. The percentages indicate the differences in the NPOD's seasonal maximum and minimum areas between El Niño and La Niña years, relative to the climatological mean.**

## 3 ENSO-induced variation in the NPOD seasonality

### 3.1 The variation in seasonal cycle of NPOD area as a response to ENSO

The amplitude of the seasonal cycle in NPOD area displays interannual variability without a significant trend during 1998–2021 (Fig. S5a), strongly correlated with ENSO (using the Niño 3.4 index, Fig. 1a). On average, the NPOD area seasonal amplitude of the El Niño years (2002, 2004, 2015, 2016 and 2019) is ~ 16% larger than for the La Niña years (1999, 2000, 2007, 2008 and 2011).

ENSO drives the variation in NPOD area seasonal cycle via its effect on the seasonal maximum of NPOD area, with little impact on the seasonal minimum of NPOD area (Fig. 1b). Even when we use the Niño 3.4 index in January, which corresponds to the month of NPOD area minimum, its correlation with the seasonal minimum in NPOD area remains insignificant (Fig. S5b). Geographically, the enhanced seasonal cycle in NPOD area in El Niño years is mainly caused by the NPOD expansion to the southeast and northeast in boreal summer when the NPOD area reaches its seasonal maximum (green lines in Fig. 2). Meanwhile, the NPOD area in boreal winter doesn't change much between El Niño and La Nina conditions, with comparably low values in both ENSO phases (grey lines in Fig. 2). Therefore, the interannual variation in the seasonal amplitude of NPOD-area shows dependence on the maximal area in summer, and this variation can be linked to ENSO.

Previous studies have reported the fast expansion of NPOD over the past two decades based on annual-mean data (Boyce et al., 2014; Meng et al., 2021), but our findings show that the expansion rate is season-dependent. As shown in Fig. 3, the NPOD expansion rate in summer, $2.7 \times 10^5$ km$^2$ year$^{-1}$, is considerably larger than either in winter, $0.7 \times 10^5$ km$^2$ year$^{-1}$, or for the annual-mean rate of $1.35 \times 10^5$ km$^2$ year$^{-1}$ (Meng et al., 2021). Although the pronounced NPOD expansion in summer may





influence the NPOD-area seasonal cycle, the trend of NPOD-area seasonal amplitude from 1998 to 2021 is not statistically significant in the presence of ENSO (Fig. S5a). Therefore, longer timescales of observations may be needed to detect a

significant change in NPOD seasonality (Henson et al., 2010; Tian and Zhang, 2024). Moreover, the interannual variation of the NPOD area, represented by its standard deviation (Fig. S6), is relatively large in summer, which further explains why the interannual variation of NPOD-area seasonal cycle depends on the summer maximal NPOD area.

### 3.2 ENSO-induced variations in ocean physics and their impact on NPOD area

To investigate the influence of ENSO on NPOD seasonality, we study the coherence between the Niño 3.4 index and various

physical parameters, averaged within the NPOD, that represent key processes influencing NPOD area (SST, MLD, upward nutrient flux to ocean surface, temperature gradient and density gradient over 5 to 55 m depth; Fig. 1). All data are averaged over the boreal summer half year (April–September), which we have shown to determine the interannual variation of the NPOD area seasonal amplitude. In El Niño years, higher SSTs and the associated increased vertical temperature gradient between 5 to 55 m lead to a strengthened thermal stratification (Fig. 1c, d). Accordingly, thermal stratification in the upper

ocean shoals the MLD and inhibits upward nutrient transport (Fig. 1e), finally leading to Chl-a concentration reduction and NPOD area expansion in EI Niño years. Therefore, the ENSO-related SST variations represent a dominant indirect ocean thermal effect on the NPOD area in the summer half year and thus on the amplitude of the NPOD area seasonal cycle.

Furthermore, ENSO can impact the NPOD area seasonal maximum by regulating equatorial upwelling. In El Niño years, suppression of equatorial upwelling and a decrease of westward advection of nutrient-rich water result in Chl-a decline in the

eastern equatorial Pacific (Fig. S7b, c) and NPOD expansion in its southern region (Fig. 2). As a result, ENSO signals can only regulate the interannual variation of NPOD area in summer when NPOD expands to its seasonal maximum and reaches the equatorial region (Fig. 2).

However, this summer-pronounced ENSO impact on NPOD is inconsistent with existing evidence, which shows that the ENSO signals have the largest impact on ocean physics in boreal winter and early spring (An and Wang, 2001). Here, one

possible reason is that ENSO-induced upwelling and SST anomalies persist through the spring and summer (Jacox et al., 2015; Wu and Kirtman, 2005). This is combined with the constraint that NPOD area can only expand in summer to the central and eastern equatorial Pacific where there are significant ENSO signals (Fig. 2).

Notably, Chl-a concentrations show low interannual variability in the western and northwestern regions of the NPOD as a response to ENSO (Fig. 2). In the western NPOD, the boundaries of the ocean desert are constrained by the relatively

eutrophic inshore regions along the western boundary, where Chl-a distributions exhibit negligible interannual variations. In the northwestern NPOD, the interannual variability in ocean desert boundaries, especially in summer (Fig. 2), is governed by the Kuroshio extension system, characterized by its pronounced eddy activity and high Chl-a concentrations (Itoh et al., 2015). The ocean dynamics and Chl-a level of the Kuroshio extension are predominantly regulated by low-frequency climate variability such as PDO (Lin et al., 2014). Therefore, the western and northwestern NPOD demonstrate insignificant

interannual variability in response to ENSO signal.



### 3.3 Impact of ENSO on Chlorophyll-a seasonal bloom in NPOD

We have also identified a connection between ENSO and Chl-a seasonality within the NPOD, as a coinciding phenomenon along with the variations in NPOD area seasonality. Here, Chl-a seasonality is characterized by the timing of its bloom, which is defined by the time when Chl-a reaches its annual peak (Fig. 4). In most NPOD regions, Chl-a blooms in the winter

half year(October–March) (Fig. 4a), which is consistent with weaker ocean stratification in winter. However, in the central area of the NPOD, Chl-a blooms in the summer half year (April–September), more specifically in May–July. After cataloguing Chl-a seasonal cycles in El Niño and La Niña years, we find that the summer Chl-a bloom in the central NPOD only occurs in La Niña years (Fig. 4c). During La Niña conditions, the northeasterly wind anomalies north of the equator in summer leads to stronger zonal wind stress and wind stress curl (Chow et al., 2019; Feng et al., 2020) in the central NPOD

area (Fig. S8d). Therefore, nutrients can be transported to the surface ocean during the pre-bloom period due to enhanced wind-induced deep mixing and horizonal nutrient transport (Toyoda and Okamoto, 2017; Wilson et al., 2013), finally leading to summer Chl-a bloom (Fig. S8b). Previous studies have attributed some Chl-a bloom events observed in the oligotrophic Pacific to nitrogen fixation (Villareal et al., 2012; Wilson and Qiu, 2008). However, the location of the Chl-a blooms in these studies is situated to the east of the Chl-a blooms in Fig. 4a. Moreover, previous research using cruise data has suggested an

insignificant impact of nitrogen fixation on summer Chl-a blooms in the NPOD region (Villareal et al., 2011). Therefore, nitrogen fixation is unlikely to be the main cause of the summer Chl-a bloom in this study. Overall, our results suggest that the two key indicators of NPOD seasonally, i.e. NPOD area seasonal amplitude and Chl-a bloom, can be both linked to ENSO via equatorial upwelling, vertical stratification, wind-induced horizontal transport and mixing processes.

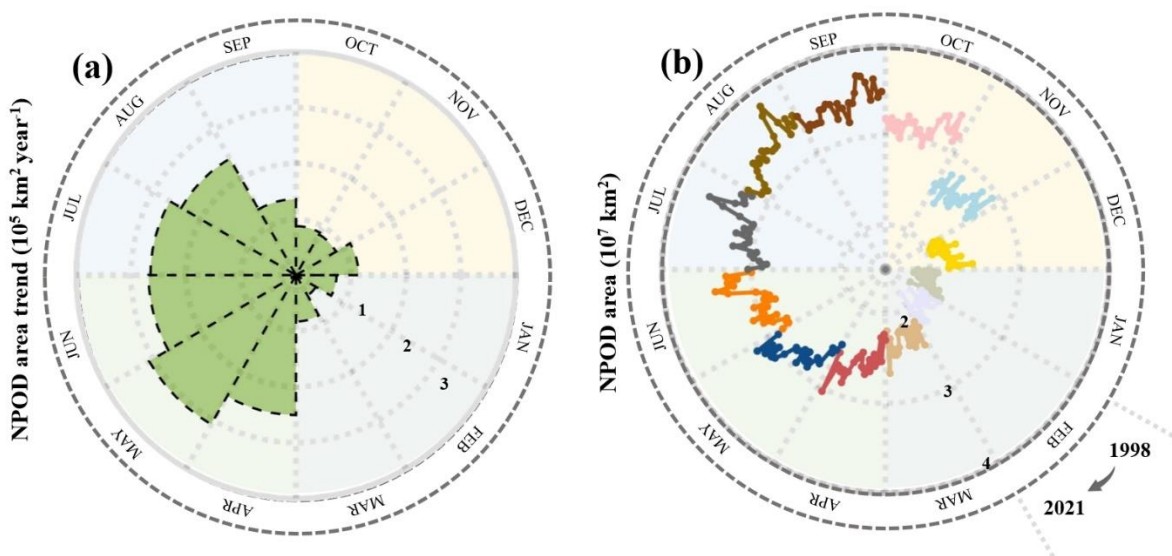

**Figure 3. Interannual variations of NPOD area in different months. (a) Trends of NPOD area expansion in 1998–2021 for specific month, according to the linear-fitted regressions of NPOD-area time series as shown in (b). (b) Time series of NPOD area in 1998-2021 for specific month.**





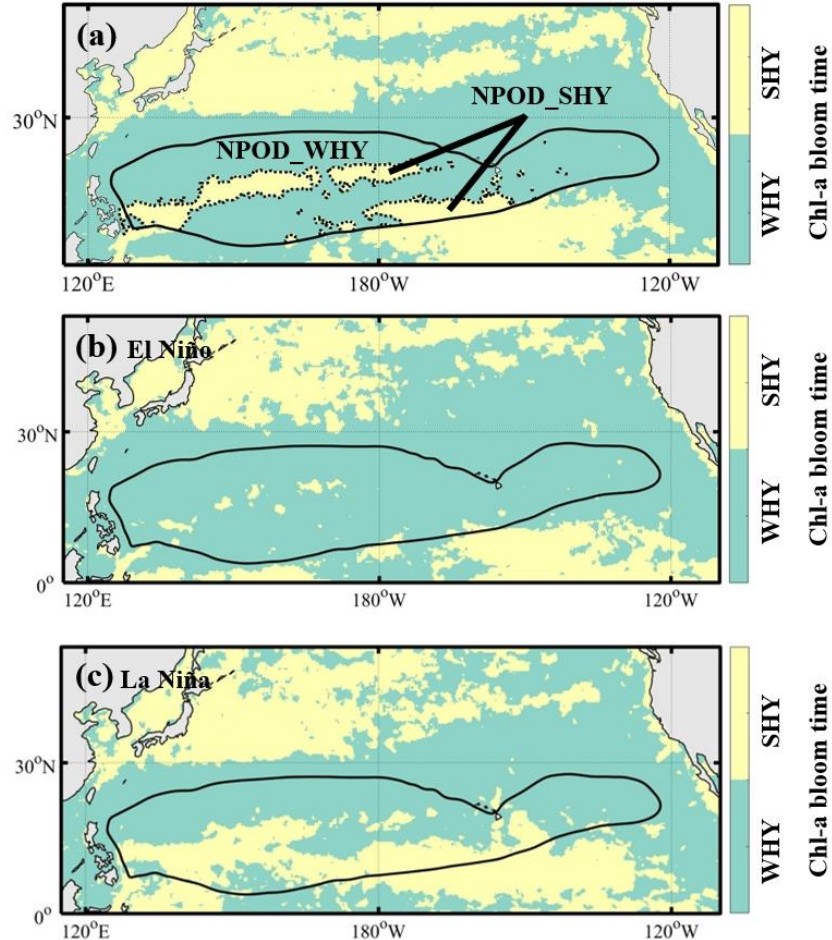

**Figure 4. Chl-a bloom time in North Pacific. (a) NPOD region is categorized to NPOD_WHY (WHY for Winter Half Year,**
**October–March) and NPOD_SHY (SHY for Summer Half Year, April–September) based on the climatological Chl-a bloom time.**
**(b-c) Chl-a bloom time in El Niño years (b) and La Niña years (c).**




## 4 Projection of the NPOD seasonality during the 21st century

### 4.1 Predicting NPOD seasonality with ENN model

To understand the response of the NPOD seasonality to future climate change, an ENN model driven by CMIP5 multi-model mean output is used to predict the amplitude of the NPOD area seasonal cycle during the 21st century. Based on a sensitivity test of ENN performance with different input configurations (Table S3), we select SST, wind stress curl and solar radiation as the optimal input data to the ENN. The selection of these variables is also in line with the discussion in Sect. 3 and the findings of Meng et al. (2021), which specifically highlight the determining impact of SST variations on ocean stratification and NPOD area.

We evaluate the ENN-predicted variations in amplitude of seasonal cycle in NPOD-area, as well as the NPOD area maximum and minimum, from 2006 to 2100 (Fig. 5). During the first half of the century (2006–2048), both the summer maximum and the winter minimum of NPOD area are predicted to expand (Fig. 5b) due to higher SST and thus enhanced upper ocean stratification (Yamaguchi and Suga, 2019). This results in a relatively gradual change in the amplitude of the NPOD area seasonal cycle (Fig. 5a), largely consistent with the insignificant trend in observations (Fig. S5a). However, a notable shift occurs in the latter half of the century (2049–2100), where the expansion of the NPOD area seasonal maximum comes to a halt, while the expansion of the seasonal minimum continues. This is likely because the expanded boundary of the NPOD maximal area is affected by other dynamical processes, such as eddy-induced mixing and upwelling at the margins of subtropical gyre (Barber and Chavez, 1983; Pennington et al., 2006), and confined in the more stably stratified subtropical gyre which prevents further expansion of the NPOD area maximum. Overall, the rate at which the seasonal minimal NPOD area expands is significantly higher than the seasonal maximal area, especially in the second half of the century, leading to a decreasing trend of the NPOD area seasonal amplitude (Fig. 5a).

### 4.2 Comparison between ENN and ESMs predictions

Although ESMs have challenges in simulating the climatological mean NPOD area (Fig. S2c), its long-term trend predicted by ESMs can still provide supporting evidence for our findings. There is considerable model-to-model variability in the trends of NPOD area and its seasonal amplitude (Fig. 6). However, the majority of models and the multi-model average, indicate a declining trend of the NPOD area seasonal amplitude from 2006 to 2100, qualitatively in agreement with the ENN projection.

While both ENN and ESMs indicate a decreasing trend in the amplitude of the NPOD area seasonal cycle, the rate of $1.7 \times 10^4$ $km^2$ $year^{-1}$ predicted by the ESMs is slower than $3.7 \times 10^4$ $km^2$ $year^{-1}$ predicted by the ENN (Fig. 6). This discrepancy can be attributed to the higher rate of expansion in the NPOD area seasonal maximum in ESMs (Fig. 6). The ENN predicts that the summer maximum ceases to increase after 2049 (Fig. 5b), which is not shown by the ESM projections (Fig. S9). This likely arises from the inaccurate simulation of the North Pacific oligotrophic ocean in ESMs. The constraints on expansion of the NPOD area seasonal maximum are not represented by ESMs in the present day (Fig. S2c), so these models will not be

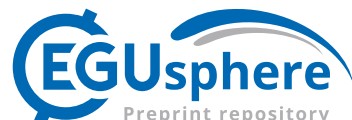

limited in their future expansion of the summer maximum, leading to a weaker decline in the seasonal amplitude in the ESMs. Nevertheless, the qualitative agreement between the ESMs and ENN projections gives us confidence in the ENN, which relies on a black box model where the underlying mechanisms are not observable.

Here, we further examine whether ESMs can capture the impact of ENSO on the NPOD area seasonal amplitude, as proposed in Sect. 3. While previous research indicates the difficulty of ESMs in simulating ENSO processes (Bellenger et al., 2014), two models, CMCC-CESM and GISS-E2-H-CC, reveal a realistic and significant correlation between ENSO and the NPOD area seasonal amplitude in their historical (Fig. S10a, b) and future (Fig. S10c, d) simulations. These two models simulate more pronounced declining trends in the NPOD area seasonal amplitude than the multi-model mean (Fig. 6), potentially indicating a physical link between ENSO and the simulated trends in NPOD area. Moreover, in the hindcast and forecast simulations, ENSO is significantly correlated with the time series of the NPOD area maximum in summer (Fig. S10e, f, g, h), in agreement with our observational findings (Fig. 1b).

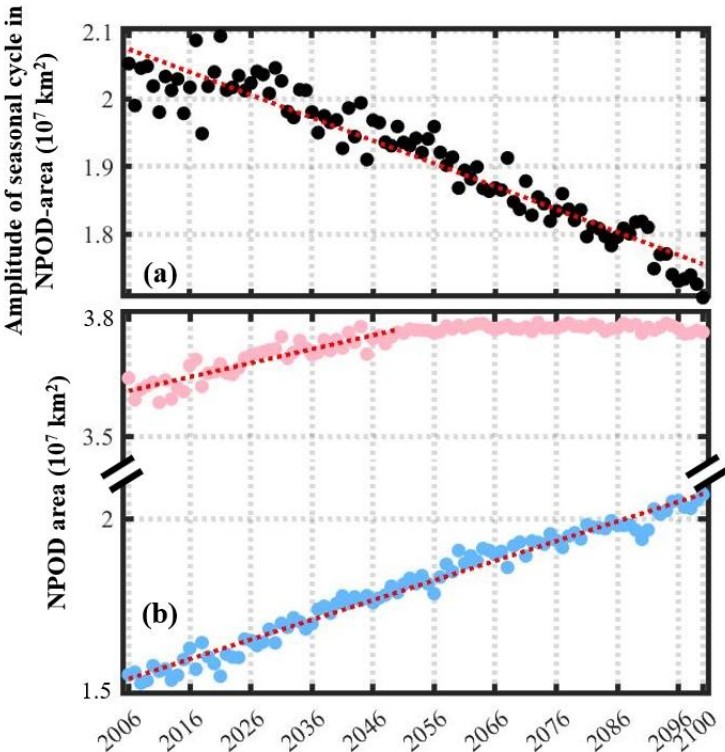

**Figure 5. Future projections of NPOD area seasonal cycle by ENN. (a-b) Time series of the seasonal amplitude (black), maximum (pink) and minimum (blue) of NPOD area during 2006–2100 predicted by ENN.**





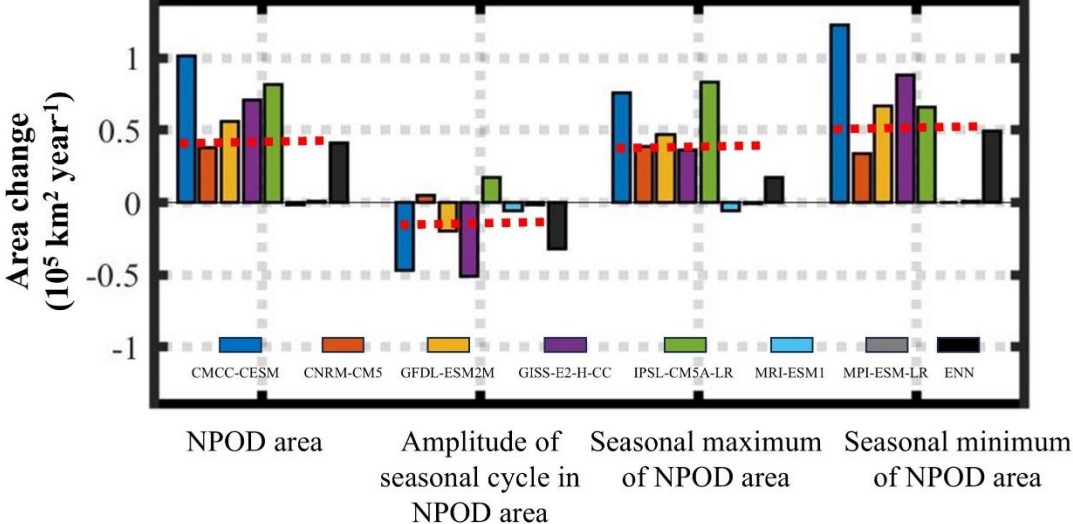

**Figure 6. Comparison of future projections of NPOD area seasonal cycle between ESMs and ENN. Variations in NPOD area, amplitude of seasonal cycle in NPOD area, NPOD area seasonal maximum and minimum during 2006–2100 projected by seven ESMs in RCP 8.5 scenario and ENN (black bar). The red dashed lines represent the average across multiple ESMs.**

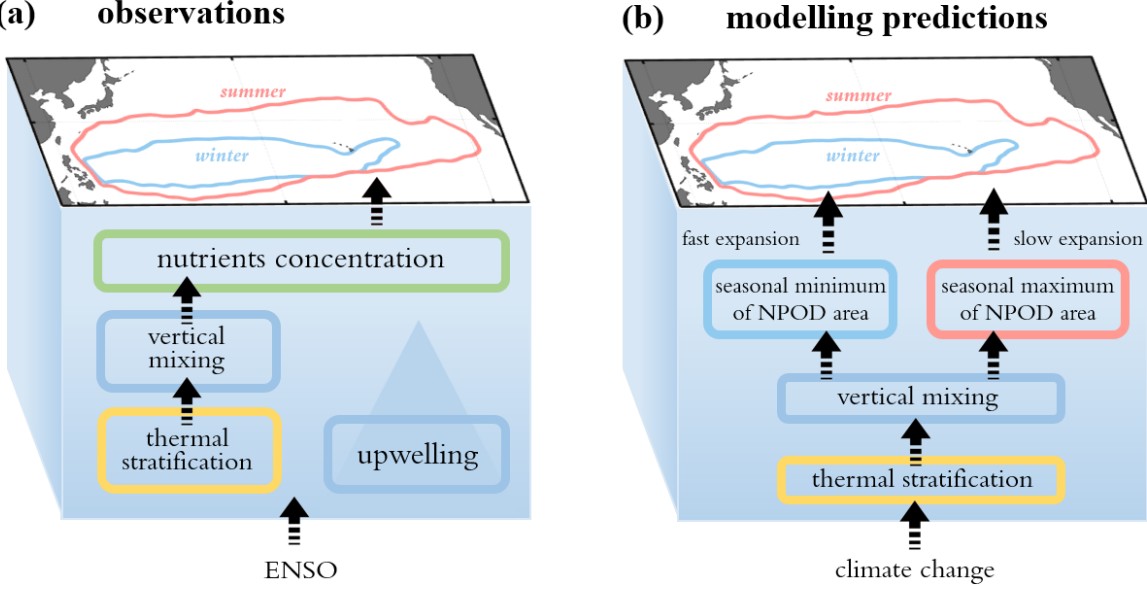

**Figure 7. Schematics of the response of NPOD area seasonal cycle to climate processes in observations (a) and modelling predictions (b). Red curves in the geographical figures represent the seasonal maximum of NPOD area in summer, and blue curves represent the seasonal minimum of NPOD area in winter.**



## 5 Conclusion

In our results, the amplitude of the seasonal cycle in NPOD area shows interannual-scale variability without a significant
trend during 1998–2021, aligned with ENSO, and the variation of the NPOD area seasonal amplitude depends on the
magnitude of the summer maximum. The impact of ENSO on the NPOD area seasonal cycle is summarized by a schematic
diagram in Fig. 7a. In El Niño years, weak equatorial upwelling and enhanced thermal stratification reduce the vertical
transfer of nutrients, finally leading to the pronounced summer expansion of NPOD area. Therefore, in the present day, we
emphasize the dominant role of ENSO-induced ocean physics on NPOD seasonality represented by the NPOD area seasonal
amplitude. The variation of NPOD seasonality is also linked to biogeochemical processes, such as phytoplankton
photoacclimation responses, which are suggested to alter phytoplankton physiology and thus Chl-a concentration in
subtropical gyres (Behrenfeld et al., 2016; Dai et al., 2023; Lewandowska et al., 2014). Therefore, future work should aim to
include additional datasets (e.g. in situ or experimental data) and biogeochemistry models for isolating the biogeochemical
impact on NPOD separately from ocean physics.

Based on an ENN projection driven by CMIP5 model output, the NPOD-area seasonal amplitude exhibits a decreasing trend
over the next century due to the faster expansion of the NPOD minimum area in winter compared to the summer maximum
area (Fig. 7b). After the middle of the 21st century, the expansion of summer NPOD area comes to a halt that is constrained
by the boundaries of the inshore North Pacific Ocean. This constraint is absent in the CMIP5 ESMs, which consistently
simulate the NPOD to extend far further than is observed in the present day.

In observations and future model projections, the interannual variation of NPOD area is different between seasons, setting
our study apart from previous research using either annual-mean (Boyce et al., 2014; Jena et al., 2013) or deseasonalized
data (Meng et al., 2021). Specifically, based on satellite observations (1998–2021), the interannual variation of the NPOD
area depends on the summer NPOD-area maximum, as the response to climate variability. However, for future projections
(2006–2100), the change of NPOD area is determined by the winter NPOD-area minimum, reflecting the response to
anthropogenic climate change. Therefore, it is essential for future analyses of NPOD expansion to consider this seasonal
difference, as the varying expansion rates of NPOD in different seasons have significant implications for the season-
dependent fisheries (Muñiz et al., 2021) and the Pacific ecosystems (Bidigare et al., 2009; Yoo et al., 2008).

NPOD seasonality is influenced by both interannual climate variability and long-term climate change, with the dominant
process determined by the time scales. On a decadal scale, the PDO and North Pacific Gyre Oscillation (NPGO) may also
impact NPOD seasonality by altering the intensity of the gyre circulations and nutrients transport (Di Lorenzo et al., 2008;
Meng et al., 2021). Although progress has been made in understanding the relationships between NPOD variations and
climate processes, there is still a challenge for future studies to separate the contribution of contemporary climate variability
and anthropogenic climate change. Such a partitioning will significantly refine our understanding of ocean ecosystem
responses to climate.

## Code availability

The ENN code and the code used for generating the figures are available in the online repository Zenodo and can be accessed via the following link: https://zenodo.org/records/14632256 (Meng et al., 2025).

## Data availability

The data used to generate the figures are available in the Zenodo online repository at https://zenodo.org/records/14632256 (Meng et al., 2025). Detailed source data and the corresponding link are provided in Text S2.

## Supplement

The supplement is available on the BG submission system

## Author contributions

SM designed the methodology, developed the software, conducted the analysis and investigation, and prepared the manuscript with contributions from all co-authors. Xun Gong contributed to validation, analysis, and resources. BW and MJ reviewed and edited the manuscript. XD supported formal analysis, while Xiang Gong contributed to the methodology. MG handled data curation. HG oversaw the project, provided funding, and guided the research.

## Competing interests

The authors declare that they have no conflict of interest.

## Acknowledgments

We thank NASA OBPG for providing satellite data. We acknowledge the Working Group on Coupled Modelling (WGCM) and the climate modelling groups (listed in Table S2) for their model output.

## Financial support

This work is funded by the NSFC-Shandong Joint Fund (U1906215); Ministry of Science and Technology of the People's Republic of China (2019YFE0125000); National Nature Science Foundation of China (41876125); State Key Laboratory of Biogeology and Environmental Geology, China University of Geosciences (GKZ22Y656); Jinan Science and Technology Bureau (202228034).



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
