# Peer review of "Seasonality of the North Pacific Ocean Desert area in the past two decades and a modelling perspective for the 21st century"

_EGUsphere, 2025_

## Author Comment (AC1)

We thank the two reviewers for their constructive comments on our work which have improved the manuscript. In this version, we have

1)included phytoplankton carbon data and calculated light conditions within the mixed layer, and further discussed how El Niño - Southern Oscillation (ENSO) modifies light availability in the mixed layer, thereby leading to phytoplankton photoacclimation responses and affecting the variation of the North Pacific oligotrophic ocean gyre.

2)redefined the key study region, and analyzed how chlorophyll variations in this region alter the interannual variability of the North Pacific oligotrophic ocean gyre.

3)provided a more balanced comparison between the Earth System Models (ESMs) and the Elman Neural Network (ENN) model, highlighting that the latter does not account for dynamical and biological processes, such as phytoplankton growth and grazing.

4)revised several figures for clarity.

Please note that in the responses to the comments below, reviewer comments are in green and our responses are in black.

**List of Responses**

**To reviewer #1:**

**1-1.** I think it would be beneficial for the science community to avoid using the term 'ocean desert'. The oligotrophic central ocean gyres are not comparable to deserts on land. The gyres are actually very biologically active and their water column net primary production is not much lower than, for example, mesotrophic systems. I recommend replacing 'ocean desert' with 'oligotrophic ocean gyre'.

Reply: Thank you for the suggestion. We agree with the comment and have replaced "North Pacific ocean desert" with "North Pacific oligotrophic ocean gyre (NPOG)" throughout the manuscript.

**1-2-1.** One of my primary concerns with this manuscript is that the analysis is based on spatial-temporal changes in surface chlorophyll concentration, and these are interpreted as reflecting phytoplankton biomass. Chlorophyll concentration, however, reflects both phytoplankton biomass and physiology (i.e., Chl:C) and the latter element reflects both nutrient availability and mixed layer light conditions. Distinguishing these factors controlling chlorophyll concentration is important as it can fundamentally impact the interpretation of observations. For example, it should be assumed a priori that chlorophyll concentration will decrease in

response to a shoaling of the mixed layer and/or increasing incident sunlight (even in the total absence of change in nutrient availability or phytoplankton biomass) simply because phytoplankton will adjust Chl:C in response to changing mixed layer light levels (i.e., photoacclimation).

Reply: We agree with the reviewer's comment and have added additional data, analyses, and discussion to address this issue.

We first made a clear distinction among surface chlorophyll concentration, phytoplankton biomass and primary productivity in the revised manuscript. To achieve this, we incorporated (1) particulate organic carbon (POC) data to calculate phytoplankton carbon and estimate phytoplankton biomass based on the methods in section 2.3 of Behrenfeld et al. (2005), as suggested by the reviewer, and calculate Chl:C ratios. (2) mixed layer light availability, following the function in section 2.1 of Behrenfeld et al. (2005), to help disentangle the effects of biomass changes and photoacclimation on chlorophyll concentration.

Second, in the Introduction we now state the assumption that, even if nutrient supply and phytoplankton biomass remain unchanged, a shallower mixed layer or higher surface irradiance is expected to lead to a decrease in chlorophyll concentration due to phytoplankton adjusting their pigment content in response to light conditions (photoacclimation) (Behrenfeld et al., 2005, 2016). We also added an analysis to show that changes in the NPOG area may be driven by both nutrient availability and photoacclimation effects (see below).

**1-2-2.** One can easily envision that the strong seasonal cycle in surface chlorophyll concentration reported in this manuscript is entirely due to this photoacclimation response and may have nothing to do with changes in phytoplankton biomass or nutrient vertical transport (see for example figure 2 in Behrenfeld et al. 2005 GLOBAL BIOGEOCHEMICAL CYCLES, VOL. 19, GB1006, doi:10.1029/2004GB002299). It can also be easily envisioned that the observed relationships between chlorophyll concentration and ENSO cycles likewise primarily reflects changes in mixed layer light levels.

Reply: We thank the reviewer for this comment. We agree that in the NPOG, temporal variations in chlorophyll and phytoplankton biomass result from a combination of biological and physical factors. Following the reviewer's suggestion, we have calculated mixed layer light availability (Behrenfeld et al., 2005) and incorporated a corresponding discussion in the revised manuscript.

In response to 1–5 comments, we have redefined our study region as the area bounded by the

interannual maximum and minimum extent of the seasonal NPOG area maximum (between the yellow and blue lines in Figure R1). Within this region, the surface chlorophyll variations largely determine the interannual variability of the NPOG seasonal maximum area (main focus of this study). We then discuss that the interannual variations of chlorophyll in this region may be influenced by two main factors:

ENSO-driven changes in the vertical thermal structure can modify nutrient availability, thereby affecting phytoplankton biomass and chlorophyll. Within the study region (Figure R1), ENSO-related changes in mixed layer depth (MLD) (Figure R2a), chlorophyll, and phytoplankton carbon are significantly correlated (Figures R2b and R2c). Spatially, the correlation between chlorophyll and phytoplankton carbon is also significant in the study region (Figure R1). Based on these results, we now apply the vertical nutrient flux calculation (originally shown in Figure 1 in the main text) to the newly defined study region in order to examine how ENSO affects thermal structure, MLD, nutrient transport, phytoplankton biomass and chlorophyll.

Even though chlorophyll and phytoplankton carbon are closely related in this region, our calculation of mixed layer light levels indicates that ENSO can also influence MLD and thus light conditions, inducing a photoacclimation response in phytoplankton that changes the Chl:C ratio and consequently the surface chlorophyll concentration (Figure R3). We have further excluded the possibility that ENSO influences chlorophyll through changes in cloud cover and ocean surface irradiance.

We thank the reviewer again for the suggestion and will incorporate the above discussion and figures into the revised manuscript.

[Figure]

**Figure R1.** Correlation coefficients between summer (April - September) surface chlorophyll and phytoplankton carbon over multiple years. The light yellow line indicates the largest interannual extent of the NPOG seasonal maximum area, and the light blue line indicates the

smallest extent.

[Figure]

**Figure R2.** Effects of ENSO on chlorophyll and phytoplankton carbon via vertical mixing. Interannual variations of the mixed layer are correlated with (a) the Niño 3.4 index, (b) surface chlorophyll concentration, and (c) phytoplankton carbon. All variables are averaged in summer half year (April-September) over the region between the yellow and blue lines shown in Figure R1.

[Figure]

**Figure R3.** Interannual variations of mixed layer light levels are correlated with the Chlorophyll-to-carbon ratio (Chl:c). All variables are averaged in summer half year (April-September) over the region between the yellow and blue lines shown in Figure R1.

**1-2-3.** It is worthwhile noting here that the ENN used in this model includes solar radiation as a primary input (i.e., photoacclimation, not nutrient stress) and that the other two inputs (SST and wind stress curl) are also linked to variations in mixed layer light levels.

We agree that the interannual variability of chlorophyll is driven by both nutrient availability and light conditions, as discussed above. Correspondingly, the ENN results also identify sea surface temperature (SST, indicating stratification) and light as the primary controlling factors, which are consistent with our findings. We note that vertical nutrient fluxes cannot be directly included as a comparable factor here, since they are not available in most of the ESMs. We have added this discussion and also clarified in the revised text that SST and wind stress curl are linked to variations in mixed layer light levels.

**1-2-4.** Unlike the photoacclimation response, it cannot be assumed a priori that mixed layer shoaling will result in a decrease in chlorophyll concentration due to a reduction in vertical nutrient transport (see for example: Lozier et al. 2011, GEOPHYSICAL RESEARCH LETTERS, VOL. 38, L18609, doi:10.1029/2011GL049414).

We agree with the reviewer that MLD cannot directly represent vertical nutrient transport. Therefore, in our study we used the K-profile parameterization (KPP) model, which accounts for thermal and salinity stratification as well as horizontal and vertical current velocities, but is independent of MLD, to estimate vertical nutrient fluxes (Figure 1e in the main text).

We also thank the reviewer for recommending Lozier et al. (2011). That study shows that wind energy in summer is often insufficient to sustain upper ocean mixing, which is a key factor driving stratification in the North Atlantic subtropical gyre. Following this insight, we emphasized not only the role of ocean stratification changes but also the potential impact of wind stress–induced mixing, which can locally increase summer chlorophyll in parts of the subtropical gyre (Figure S8 in the supporting information).

Lozier et al. (2011), based on a 10-year dataset, reported no correlation between stratification and chlorophyll. By contrast, our analysis does reveal such a relationship, which may be due to (1) the longer time period considered in our study and (2) differences in the selected study region. In addition, we now emphasize that changes in chlorophyll are influenced not only by nutrient mixing but also by phytoplankton photoacclimation processes.

**1-2-5.** The accurate interpretation of mechanism driving observed surface chlorophyll concentrations is important throughout this manuscript. For example, a decrease in chlorophyll

due to photoacclimation to higher mixed layer light levels is expected to be associated with either unchanged or increased primary production, not a decrease.

**Reply:** We agree with the reviewer's comment. In response, we have revised the manuscript to explicitly distinguish between changes in chlorophyll concentration, phytoplankton biomass, and primary production. We clarified that a decrease in surface chlorophyll results from photoacclimation due to enhanced mixed layer light levels, which can also increase light penetration into deeper layers (Manizza et al., 2005; Meng et al., 2024), thereby likely leading to higher vertically integrated phytoplankton biomass and primary production.

**1-2-6.** Another example is that a photoacclimation-based chlorophyll response makes the evaluation of phytoplankton 'blooms' in the oligotrophic north pacific gyre very questionable. The term 'bloom' is usually associated with a significant change in phytoplankton biomass, not a seasonal change in light driven (or nutrient-driven for that matter) change in Chl:C. Thus, without carefully distinguishing light-, nutrient-, and biomass-driven changes in chlorophyll concentration, the section of the manuscript regarding bloom properties is compromised.

Reply: We agree with the concern raised. Accordingly, we have revised the manuscript by replacing the term 'bloom time' with 'chlorophyll peak time', since our definition was based on the seasonal maximum in chlorophyll concentration. In addition, we have expanded our analysis to include the relationships between chlorophyll peak time, mixed layer light availability, phytoplankton carbon and Chl concentration (see below).

**1-2-7.** It may also be noted here that the more common NPOD_WHY feature shown in figure 4 is consistent with photoacclimation to winter minima in mixed layer light levels and that the less common NPOD_SHY also corresponds (according to the authors) to regions where summer mixed layer depths are high (i.e., lower light).

Reply: We thank the reviewer for this comment. In response, we have added a discussion to clarify the mechanisms underlying these features. For the NPOG_WHY region, we note that the winter maximum in chlorophyll arises from the combined effects of deeper mixed layers, increased nutrient availability, and reduced light within the mixed layer that elevates the Chl:C ratio. In the NPOG_SHY region, which is the main focus of our study, we find that variations in mixed layer light availability are strongly correlated with changes in the Chl:C ratio, and thus likely represent the dominant driver of the summer chlorophyll maximum (Figure R4). At the same time, we also acknowledge the potential role of wind-driven mixing and nutrient supply, as noted in the original manuscript (Figure S8 in the supporting information).

[Figure]

**Figure R4.** Interannual variations of mixed layer light levels are correlated with the Chlorophyll-to-carbon ratio (Chl:c). All variables are averaged in summer half year (April-September) over the NPOG_SHY region as shown in Figure 4a.

**1-2-8.** The importance of light- versus nutrient-driven chlorophyll changes also compromises the validity of the conceptual model presented in figure 7.

Reply: We have revised Figure 7 (see Figure R5) to better reflect the distinction between light- and nutrient-driven changes in chlorophyll. Specifically, on the left panel, which illustrates the influence of ENSO on the seasonal cycle of the NPOG area, we have explicitly included the role of light availability and its impact on chlorophyll through photoacclimation (i.e., changes in Chl:C). On the right panel, which presents model-based projections under climate change, we now highlight three key drivers, temperature, light, and wind curl, recognized by ENN model as the main factors influencing NPOG area variability.

[Figure]

**Figure R5.** Schematics of the response of NPOG area seasonal cycle to climate processes in observations (a) and modelling predictions (b). Red curves in the geographical figures represent

the seasonal maximum of NPOG area in summer, and blue curves represent the seasonal minimum of NPOG area in winter.

**1-3.** It is noteworthy that a decrease in surface chlorophyll concentration will correspond to a decrease in mixed layer light attenuation coefficients, causing submixed layer light levels to increase and thus submixed layer primary production to increase, again questioning the quantitative significance of surface chlorophyll concentration changes to overall productivity.

We agree with this comment from reviewer. As we noted in our reply to comment 1-2-5, we clarified that a decrease in surface chlorophyll results from photoacclimation due to enhanced mixed layer light levels, which can also increase light penetration into deeper layers (Manizza et al., 2005; Meng et al., 2024), thereby likely leading to higher vertically integrated phytoplankton biomass and primary production.

**1-4.** Figure 3 provides an interesting analysis of temporal trends in chlorophyll concentration, but it seems it would be useful to also show an overall time series of these trends. Figure 3b does this to a degree in a monthly-resolved manner, but there is no indication in this panel which of the monthly trends are statistically significant.

Reply: We thank the reviewer for this suggestion. In response, we plotted the overall time series of the NPOG area to provide a complete view of its seasonal and interannual variability. We then selected several representative months for multi-year linear fitting, which shows that the long-term area expansion trend is stronger in summer than in winter (Figure R6). In addition, in Figure R7 (revised from Figure 3b), the months with significant interannual trends at the 0.05 level are highlighted by yellow lines to clearly indicate statistical significance.

[Figure]

**Figure R6.** Temporal variations in the NPOG area in 1998-2021. Linear regressions for the months of June, July, August, December, January and February of each year are shown by colored dashed lines.

[Figure]

**Figure R7.** Interannual variations of NPOG area in different months. (a) Trends of NPOG area expansion in 1998–2021 for specific months, derived from the linear-fitted regressions of NPOG-area time series as shown in (b). (b) Time series of NPOG area from 1998 to 2021 for specific months. The yellow lines in (b) represent the linear regressions that pass the significance test at the 0.05 level.

**1-5.** In figure 2 and as discussed in the text, changes in summer NPOD area between El Nino and La Nina conditions are not widespread but rather primarily isolated to the two regions indicated in figure 2. It is therefore not clear to me why the influence of ENSO was evaluated based on physical properties averaged over the entire NPOD (line 200). Why wasn't this analysis focused on physical changes only in the areas where ENSO effects are seen? If the

assessed changes in physical properties are representative of the entire NPOD, why are there no changes in chlorophyll concentration observed over the entire region?

We agree with the reviewer that averaging physical properties over the entire NPOG region does not capture the spatially confined ENSO signals shown in Figure 2. Our study, however, specifically focuses on the interannual variability of the seasonal maximum NPOG area. To investigate this, we defined a main study region bounded by the interannual maximum and minimum extent of the seasonal maximum NPOG area (between the yellow and blue lines in Figure R1). Chlorophyll variability within this region directly determines the interannual changes in the seasonal maximum NPOG area.

Within this region, we find that ENSO signals are closely linked to the interannual variability of chlorophyll, mainly through the modulation of nutrient availability and light conditions (Figures R2 and R3). Additionally, we also discussed the influence of ENSO on chlorophyll in specific subregions within this area, for example through upwelling in the eastern Pacific (Figure S7) and via Kuroshio-related processes along the northwestern NPOG boundary (main text).

**1-6.** It seems to me that the manuscript is a bit critical of the Earth System Model results without being equally critical of the ENN results. For example, how reliable are the ENN predictions about future change when the ENN is built from hindcast data that doesn't take into account future changes in major ocean physical features (e.g., a potential northward movement in the location of the Kuroshio current) that provide critical constraints on the potential areal extent of the oligotrophic north pacific gyre? I think a more balanced evaluation of strengths and weakness of different approaches is warranted.

Reply: We thank the reviewer for this comment. We agree that a more balanced discussion of the strengths and weaknesses of the two approaches is needed. We have revised the manuscript to highlight that while ESMs provide process-based simulations, they are subject to systematic biases and model uncertainty. In contrast, the ENN approach can efficiently extract statistical relationships from historical data, but its predictions are limited by the training dataset and do not explicitly account for potential future changes in major ocean physical features (e.g., shifts in the Kuroshio Current). Furthermore, ENN operates as a "black box", making mechanistic interpretation less straightforward. These limitations are now discussed in the revised text.

Minor comments:

**1-7.** the light colored symbols and lines in figure 1 are nearly impossible to see. I suggest bolder colors. Same issue in figure 2 regarding the La Nina lines.

Reply: Thanks for the suggestion. We will modify the figures with bolder colors to improve visibility and will include the revised versions in the updated manuscript submission.

**1-8.** the black contours in figure 4 are not defined in the caption.

Reply: Thank you for pointing this out. The black contours in Figure 4 represent the multi-year mean NPOG boundary. We will add this information to the figure caption in the revised manuscript.

**To reviewer #2:**

This paper provide a bottom-up (nutrient/stratification) analysis to the dynamics of phytoplankton in the North Pacific subtropical gyre. The premise of this paper is that this region is a desert that is modulated by nutrient dynamics. This premise is wrong for many reason which I will detail below:

**2-1.** As Ed Laws has shown in his ARMS review, phytoplankton cells in this and similar (surface) area divide once a day. If this is the case, why call it a desert?

Reply: We thank the reviewer for this comment. Following the suggestion, we have replaced the term "North Pacific Ocean Desert (NPOD)" with "North Pacific oligotrophic ocean gyre (NPOG)" throughout the manuscript, also suggested by reviewer #1. In addition, we have added the discussion to clarify that, although phytoplankton biomass in this region is low, its vast area and the fact that phytoplankton cells can divide rapidly (Laws, 2013) mean that such biological activity has significant ecological impacts.

**2-2.** The designation of desert is based on surface. [chl a]. [Chl a] is a problematic biomass indicator due to photoacclimation. what about phytoplanton carbon or nitrogen, and in particular, depth integrated? Shouldn't the depth integrated value be what we look at when considering the contribution to the ecosystem rather than surface concentrations?

Reply: We agree with the reviewer that chlorophyll is not a direct measure of phytoplankton biomass. In the revised manuscript, we therefore make a clear distinction between surface chlorophyll, phytoplankton biomass, and primary productivity. In addition, we used MODIS-derived particulate organic carbon (POC) data to calculate the phytoplankton carbon together with the method described in Behrenfeld et al. (2005) to calculate light conditions within the mixed layer.

As suggested by reviewer 1 (comment 1-5), we have redefined a main study region bounded by the interannual maximum and minimum extent of the seasonal maximum NPOG area (between the yellow and blue lines in Figure R1). Chlorophyll variability within this region directly determines the interannual changes in the seasonal maximum NPOG area.

Our results show that during the MODIS observational period, ENSO is linked to the interannual variability of the summer NPOG area primarily through: (1) changing thermal stratification and thus vertical nutrient supply, which impacts chlorophyll (Figure R2b) and is

also reflected in phytoplankton carbon variability (Figure R2c), as further suggested by the nutrient flux calculation we applied to the new study region; and (2) modifying light availability within the mixed layer, leading to a photoacclimation response that changes the chlorophyll-to-carbon ratio (Figure R3). Accordingly, we will add a discussion in the manuscript on how ENSO-driven changes in MLD and light conditions influence the chlorophyll-to-carbon ratio.

We also agree that depth-integrated phytoplankton biomass and productivity are more relevant measures of ecological contribution. However, such observational data are not available on a global scale and over long time periods . To address this, we have added discussion on how surface chlorophyll can influence depth-integrated productivity, for example: a decrease in surface chlorophyl results from photoacclimation due to enhanced mixed layer light levels, which can also increase light penetration into deeper layers (Manizza et al., 2005; Meng et al., 2024), thereby likely leading to higher vertically integrated phytoplankton biomass and primary production.

[Figure]

**Figure R1.** Correlation coefficients between summer (April - September) surface chlorophyll and phytoplankton carbon over multiple years. The light yellow line indicates the largest interannual extent of the NPOG seasonal maximum area, and the light blue line indicates the smallest extent.

[Figure]

**Figure R2.** Effects of ENSO on chlorophyll and phytoplankton carbon via vertical mixing. Interannual variations of the mixed layer are correlated with (a) the Niño 3.4 index, (b) surface chlorophyll concentration, and (c) phytoplankton carbon. All variables are averaged in summer half year (April-September) over the region between the yellow and blue lines shown in Figure R1.

[Figure]

**Figure R3.** Interannual variations of mixed layer light levels are correlated with the Chlorophyll-to-carbon ratio (Chl:c). All variables are averaged in summer half year (April-September) over the region between the yellow and blue lines shown in Figure R1.

**2-3.** Phytoplankton accumulation, e.g. the change of concentration with time, is one to two order of magnitude smaller than their growth-rate, indicating that loss processes (e.g. grazing

and viruses) are just as important as growth inducing processes in the dynamics of phytoplankton. While I do understand that it is hard to study these processes, ignoring them will not help in understanding their accumulation dynamics. In the least one has to acknowledge the equal importance of these processes and the assumption models do when parametrizing them (e.g. tuning to get correctly the average chlorophyll, etc').

Reply: We thank the reviewer for raising this important issue. We fully agree that phytoplankton accumulation is the net outcome of both growth and loss processes, and that the latter (e.g., grazing) play an equally important role in regulating dynamics. While these processes are indeed difficult to quantify, we will acknowledge their importance and add the discussion below in the revised manuscript.

Biogeochemistry models within Earth System Models (ESMs) usually simplify phytoplankton loss processes by representing them through empirical parameterizations rather than mechanistic formulations. For example, grazing is often represented as a fixed fraction of phytoplankton biomass or growth, while other losses (e.g., mortality, viral lysis) are lumped into mortality terms. Such simplifications are necessary for computational efficiency, but they rely on assumptions and tuning (e.g., adjusting grazing coefficients) to reproduce observed mean chlorophyll levels.

In particular, we now clarify that the Elman Neural Network (ENN)-based simulation we present does not explicitly represent phytoplankton growth or loss processes, but instead provides an empirical reconstruction of chlorophyll variability. Accordingly, we have emphasized that the results should be interpreted as reflecting correlations between environmental drivers and chlorophyll, rather than a mechanistic representation of phytoplankton growth. We further note that future work should aim to explicitly integrate grazing and loss processes to improve mechanistic understanding of phytoplankton accumulation dynamics.

Reference:

Behrenfeld, M. J., Boss, E., Siegel, D. A., and Shea, D. M.: Carbon-based ocean productivity and phytoplankton physiology from space, Global Biogeochem Cycles, 19, 1–14, https://doi.org/10.1029/2004GB002299, 2005.

Behrenfeld, M. J., O'Malley, R. T., Boss, E. S., Westberry, T. K., Graff, J. R., Halsey, K. H., Milligan, A. J., Siegel, D. A., and Brown, M. B.: Revaluating ocean warming impacts on global phytoplankton, Nat Clim Chang, 6, 323–330, https://doi.org/10.1038/NCLIMATE2838, 2016.

Laws, E. A.: Evaluation of in situ phytoplankton growth rates: A synthesis of data from varied approaches, Ann Rev Mar Sci, 5, 247–268, https://doi.org/10.1146/annurev-marine-121211-172258, 2013.

Lozier, M. S., Dave, A. C., Palter, J. B., Gerber, L. M., and Barber, R. T.: On the relationship between stratification and primary productivity in the North Atlantic, Geophys Res Lett, 38, 2011GL049414, https://doi.org/10.1029/2011GL049414, 2011.

Manizza, M., Le Quéré, C., Watson, A. J., and Buitenhuis, E. T.: Bio-optical feedbacks among phytoplankton, upper ocean physics and sea-ice in a global model, Geophys Res Lett, 32, 1–4, https://doi.org/10.1029/2004GL020778, 2005.

Meng, S., Webber, B. G. M., Stevens, D. P., Joshi, M., Palmieri, J., and Yool, A.: Diverse Responses of Upper Ocean Temperatures to Chlorophyll-Induced Solar Absorption Across Different Coastal Upwelling Regions, Geophys Res Lett, 51, https://doi.org/10.1029/2024GL109714, 2024.